# Evaluation of Surface Roughness Parameters of HDF for Finishing under Industrial Conditions

**DOI:** 10.3390/ma15186359

**Published:** 2022-09-13

**Authors:** Milena Henke, Barbara Lis, Tomasz Krystofiak

**Affiliations:** Department of Wood Science and Thermal Techniques, Faculty of Forestry and Wood Technology, Poznan University of Life Sciences, Wojska Polskiego 28, 60-627 Poznan, Poland

**Keywords:** roughness, HDF board, sanding belt, grain sizes, feeding speed

## Abstract

One of the most important properties of the surface of wood-based panels is their roughness. This property determines the way of working with the material in the processes of gluing and surface varnishing. The aim of this study was to determine the effect of various sanding belt configurations and the feeding speed of the conveyor belt during grinding on the surface roughness of high-density fiberboards (HDF). The research material was prepared under industrial conditions. Three types of boards were selected for the tests. After grinding, the roughness parameters were measured both transversely and longitudinally relative to the grinding direction, using a Mitutoyo SJ-210 profilometer and the optical method. Based on ANOVA analysis of the data, it was found that the type of HDF boards used and the configuration of the abrasive belts had a statistically significant impact on the roughness. The samples for which the grinding process was performed with sanding belts of the highest grain size had the lowest roughness. For the amplitude roughness parameters, the direction of roughness measurement had a significant influence. These results may provide valuable guidance for the furniture industry in the preparation of HDF for furniture production.

## 1. Introduction

Due to advances of civilization and the continuous improvement of living standards, expectations of the furniture industry are continually increasing. These expectations stimulate the development of technologies for use in the wood industry. One of the most important features for the customer is the overall visual impression and the quality of the surface finish of individual furniture elements [1]. The roughness of the surface of wood and wood-based materials is one of the most important parameters affecting the appearance of final products and the processes of gluing and surface finishing [2,3]. The varnishing of smoother surfaces is more efficient, since a satisfactory effect can be achieved with smaller quantities of varnish products. Meanwhile, in the gluing process, a lower surface roughness has a positive effect on the distribution of the adhesive [4,5].

The surface roughness depends on the physical and chemical properties of the substrate, which are taken into account when evaluating a material. In the case of wood, characterization of the geometric structure is particularly difficult due to its anatomical and morphological features. It requires the analysis of many factors that influence the final result [6,7]. The roughness is influenced by the type of wood (soft or hard) and the resulting wood density and porosity (higher density correlates with lower porosity, resulting in a smoother surface), the width of the annual rings, the percentage ratio of early to late wood, the type and structure of the cells, and even the number and arrangement of tracheid and vessel elements and medullary rays. Moreover, wood has anatomical defects that change the local structure and density of the material [8,9,10,11,12,13,14]. The areas around the defects usually have lower roughness than the areas without defects [2]. Wood-based panels, especially high density fiberboards, are more homogeneous than wood itself [15].

The quality of the finish is very important in the commercial production of wood or wood-based functional items. It depends on the smoothness of the substrate [16]. It is, therefore, necessary to know the topography of the substrate that is to be finished. It is essential for furniture technology experts to be familiar with roughness and means of shaping it. By being able to select the type of abrasion and granulation during processing, it is possible to effectuate the intended profile change while also influencing the wear of the applied lacquer products, as well as the technical and optical properties of the lacquer coatings obtained from them [17].

Research on HDF (high-density fiberboards) boards has shown a strong relationship between their density and the values of roughness parameters. They exhibit better surface stability than MDF (medium density fiberboard) [18]. Other experimental studies show that the *Ra* (arithmetic average roughness) parameter of MDF, stimulated by various factors, increased on average by 0.90 to 2.36 times, compared with the particleboard of the same density [19]. The conditions of storage, variable temperature of the substrate during processing, and humidity also play a significant role in shaping the surface of wood and wood-based materials [11,20,21]. An increase in humidity has a negative impact on all measures of the surface quality of raw, laminated, and sandwich wood-based panels [19,22,23]. Moreover, surface roughness is closely related to the machining parameters. During the cutting or sanding process, the influence of such factors as the type of cutting agent, knife geometry (rake angle, clearance), marks per centimeter, tool speed, tool wear, and cutting direction (longitudinal, radial or tangential) have been reported [11,24,25]. With an increase in machining precision, lower roughness values were obtained [12].

Grinding performs a significant role in the production of furniture, where wide-belt sanders are often used. There is a widespread belief that good painting is impossible without proper sanding. Wide-belt sanders are commonly used in sanding processes. Properly selected sanding belt and machine settings, including the feeding speed, the motion of the abrasive belt, and the pressure exerted, ensure high quality and optimal grinding parameters to obtain the lowest possible roughness of the wood [26,27,28]. It should be taken into account that structural changes occurring on the surface of the substrate will include anatomical irregularities and those formed during the grinding process. Therefore, the anatomical roughness should be excluded from the assessment of the effect of sanding on the roughness of wood. This can be achieved using the Abbot curve method [29,30,31]. The variety of materials and manufactured finished products makes it necessary to look for individual solutions for sanding technology [32]. In the production of commercial elements made of wood or wood-based materials, the quality of the finish is very important. It depends, among other things, on the smoothness of the substrate. Therefore, it is necessary to know the topography of the substrate undergoing finishing. This issue is of greater importance in thin-film applications, especially in printing technologies carried out at high line speeds. The unevenness profile here plays a very important role in obtaining finishes of high aesthetic and decorative value. The shaping of the visual and functional features of varnish coatings begins when the surface is prepared for varnishing. This is a key stage in the substrate improvement technology [18,33,34,35,36,37,38,39].

Recently, there has been an increase in commercial interest in the implementation of UV varnish products, influenced by global trends, resulting from the need to take action to protect the environment and by changes in customer behavior and requirements [40,41,42,43,44,45,46]. The quality of manufactured products, and thus the finishes, must be improved to gain the acceptance of buyers. The furniture industry places very high demands on the appearance of finishes on final products while seeking to reduce costs by increasing efficiency and reducing processing time [27,32,47,48,49]. This entails the introduction of changes in the technology of product manufacture, beginning with surface preparation. This concerns the selection of abrasive materials, their grain size, and configuration of their operation, as well as the achievement of the desired accuracy of machining for the final result to be effective [39,50,51].

The various technologies of furniture production, especially the methods of substrate preparation and the applied process parameters, mean that coatings may differ in structural properties and physical parameters, including optical ones [39,52]. The introduction of any technological modifications in substrate treatment leads to changes in the quality of the finished product, as there is a close connection between the material after grinding and the coating [49,53]. The roughness must, therefore, be taken into account when applying varnish products. Decisions regarding the choice of sanding technology should be made on the basis of measurements of the roughness profile [33,52,54].

The latest trends in the furniture industry indicate the development of furniture made of sandwich panels with a honeycomb filling, the facings of which are often HDF boards [55]. Studies on the roughness of wood-based boards are published less frequently than for wood. Proper processing by sanding enables the obtaining of surfaces with the required roughness for finishing with varnish products or decorative veneers. To the authors’ knowledge, while there are many works in the existing literature related to substrate preparation by the sanding of various types of wood, MDF, or particleboards [22,33,56], studies on the roughness of HDF boards are rare [37,57].

The aim of the present study was to analyze the surface grinding process of sandwich panels with HDF facings, taking into account different HDF densities, grain sizes of sanding belts, and feeding speeds. It was assumed that these factors have a significant impact on the roughness profile parameters. The surface sanding process of HDF boards was carried out in a technologically advanced factory that has recently introduced many process innovations. In general, the dissemination of such data by companies is limited, due to the fact that each manufacturer, through its own experience, develops a grinding system that becomes a source of competitive advantage.

Due to the many possibilities of combining devices, sanding materials, and individually selected sanding parameters, it is possible to achieve high accuracy of work even at high production line speeds. In the experiment, roughness parameters were recorded using a Mitutoyo SJ-210 profilometer and the One Attension Theta optical tensiometer with a 3D topography module. Moreover, for the selected boards and sanding variants, the surface gloss was determined with a Pico Gloss 503 photoelectric gloss meter.

The influence of individual factors on the surface quality was examined using the ANOVA method. The results of this research may contribute to improvements of the surface preparation of HDF boards used in finishing processes, by providing knowledge about the influence of technological parameters on the quality of boards used in the furniture production sector.

## 2. Materials and Methods

### 2.1. Materials

The base material was a board on a frame with a honeycomb core (Figure 1). As external facings, three different types of HDF boards were used, with a nominal thickness of 2.5 mm, hereinafter referred to as A, B, and C (Table 1). The core of the board is recycled paper with a weight per area of 140 ± 5 g/m^2^ determined on the basis of ISO 536. Particleboard with a thickness of 29 mm and a density of 640 ± 10 kg/m^3^ was used in the frame construction. The claddings were bonded to the honeycomb core with a PVAC adhesive with a viscosity of 14,000 ± 300 mPas determined with a Brookfield DV2T viscometer at a processing temperature of 40 ± 0.2 °C.

### 2.2. Sample Preparation

In the first step, cellular paper, particleboard, and HDF boards were cut to appropriate sizes. Then, a frame structure was prepared, which was filled with honeycomb paper, and PVAC adhesive was applied to the HDF board. The facings were glued to the prepared structure and pressed. A sample prepared in this way, with a total thickness of 34 mm and dimensions of 1400 × 600 mm, was subjected to grinding. A Heesemann LSM8 + EA10 wide-belt sander, consisting of five grinding units and a brush to pre-clean the surface, was used for the tests. The aggregates were equipped with a sawdust suction system and an oscillating blade blowing off the abrasive belt, which meant that the dust generated during processing was removed from the abrasive surface. Six different configurations of sanding belts with corundum coating, 1370 mm wide, were selected. In three of them, one abrasive belt with grain size P150, P220, or P400 was used; the others included two belts in the following configurations: P150–P220, P220–P320, P400–P400. Each grinding sequence was performed at a sanding belt speed of 5 m/s. The pressure of the pressure beam (Heesemann CSD system) applied to the sanding belt via the graphite sliding liners was set to 30% of the aggregate pressure force scale. After grinding, the samples were transferred to a device where the surface was cleaned and deionized. Each variant was used with two different feeding speeds of the conveyor belt. The conveyor belt was equipped with a vacuum system and ensured the rectilinear movement of elements under the grinding aggregates. In total, 36 variants were prepared under production conditions, at a temperature of 21.5 ± 0.5 °C and air humidity of 36 ± 2%.

### 2.3. Roughness

The differential induction method was used for the tests, using a Mitutoyo SJ-210 portable spindle profilometer with a diamond measuring tip having a radius of 2 µm and an angle of 60°. The test was performed in accordance with the procedure contained in PN-EN ISO 4287: 1999/A1: 2010, with the following parameters:-Detector measuring force 0.75 mN;-Feed speed 0.5 mm/s;-Measuring range 5.6 mm;-Cut-off length λc = 0.25 mm.

The profilometer was calibrated every 50 measurements using a standard reference board with an *Ra* value of 1.75 µm. Twenty transverse and longitudinal measurements were made on each sample at randomly selected points on the entire surface of the board. Seven roughness parameters were identified, including amplitude parameters: arithmetic mean deviation (*Ra*), geometric average roughness (*Rq*), kurtosis of the roughness profile (*Rku*), and skewness of the roughness profile (*Rsk*); and vertical parameters: the maximum peak height of the roughness profile (*Rp*), the maximum valley depth of the roughness profile (*Rv*), and ten-point height (*Rz*) [31,58,59].

In addition, an analysis of the surface topography was performed by means of a non-invasive procedure using the OneAttension Theta optical tensiometer with a 3D topographic module (Biolin Scientific AB, Västra Frölunda, Sweden). The parameters of the measurement system are given in Table 2.

For this purpose, from among the tested variants, two boards of different densities (A and C) were selected, and their surfaces were ground using two programs with gradations of P150 or P150 and P220 and two speeds (25 and 50 m/min). The selected systems were the subject of research reported in a previous article by the same authors [60]. The boards were cut into 10 × 10 cm samples, on which ten measurements were made at randomly selected points. After completing a given test, reports of roughness parameters were generated in numerical form.

### 2.4. Gloss Measurement

The gloss was determined in accordance with the DIN 67530: 1982 and ISO 2813: 1994 standards, using a Pico Gloss 503 photoelectric camera (ERICHSEN GmbH & Co. KG, Hemer, Germany) [61,62]. Twenty measurements were made on each sample. The gloss grade for the 60° angle was classified according to the following criteria (Akzo Nobel 2022): GU < 10 matt, GU 10–35 semi-matt, 35–60 semi-gloss, GU 60–80 gloss, GU > 80 high gloss [63].

### 2.5. Data Processing

Minitab 19 software was used for statistical analysis of the test results. To determine the influence of individual factors on the surface roughness, analysis of variance (ANOVA) was performed. A main effects plot was used to present the data.

## 3. Results and Discussion

### 3.1. Profilometer Measurement Method

The normal distribution hypothesis was verified using the Ryan–Joiner test, similar to the popular Shapiro–Wilk test. At the significance level of α = 5%, the roughness profile asymmetry coefficient *Rsk* was not consistent with a normal distribution (*p*-value = 0.034). The Johnson transformation was performed for this parameter, the data became normal distribution (*p*-value = 0.801). For the remaining parameters, the data were normally distributed for raw data, and the null hypothesis that the variance of the dependent variable error was equal in all groups was accepted at the set confidence level (>0.05) [64].

ANOVA was performed. The difference between the raw data and the post-transformation results is not statistically significant. The data were assessed on the basis of four variables: HDF board type (3), gradation of abrasive belts (6), feed speed (2), and measurement direction—longitudinally or transversely to the grinding direction (Table 3).

It was found that the mean values of the amplitude parameters *(Ra*, *Rq*, *Rsk*, *Rku*) differed at the significance level α = 0.05, depending on the HDF boards used, the configuration of abrasive belts, and the direction of roughness measurement (Figure 2). The exception was that the *Rsk* and *Rku* parameters were not significantly influenced by the HDF type.

The height roughness parameters (*Rz*, *Rp*, *Rv*) were significantly influenced by two components: the type of HDF board used and the configuration of the abrasive belts. The exception here was the *Rv* parameter, for which the *p*-value was 0.939 when the analyzed factor was the configuration of abrasive belts. The direction of roughness measurement did not have a significant impact on the values of these parameters (Figure 3). The remaining factor, the conveyor belt speed, had no significant influence on either the height or amplitude parameters.

The type of HDF board used had a large influence on the obtained roughness. As stated at the outset, three boards were used, thus introducing two variables: two different manufacturers and two densities. For the *Ra* parameter, HDF board B with a density of 850 kg/m^3^ had a roughness 6.7% lower than HDF board C from the same manufacturer with a density 20 kg/m^3^ lower. This corresponds to the findings on the effect of density on the geometric structure of the wood surface cited in the first section. Board A, with a density of 850 kg/m^3^ but purchased from a second manufacturer, obtained an average *Ra* value 27.9% higher than that of board B (density 850 kg/m^3^) and 19.35% higher than that of the 830 kg/m^3^ board from the other manufacturer, which is contrary to the general trend for the effect of density. Despite significant progress in the determination of the relationship between factors affecting the properties of wood and the parameters of material processing and surface roughness, no generally applicable correlation has been established [10,56]. Confirmation of this is provided by the latest research on the roughness of veneers of various wood species, in which the lowest values of geometric structure parameters were measured on the surface of chestnut wood, compared with other species with a higher density. The conditioning and processing of veneers were carried out in the same way, which indicates the influence of other morphological features of the wood, as well as a slightly different equilibrium moisture content [2]. Attention should also be given to research on the seasonal variability of fiberboard properties depending on the processed grade and the degree of chemical degradation [65,66]. Fibers made of a mixture of Scots pine and beech chips have the lowest MDF surface roughness, and boards made of poplar, birch, and Scots pine have slightly higher values. The roughest surfaces are obtained on boards made of beech and oak fibers [56,67]. The cited studies explain the observed inconsistency in the trend for roughness to increase along with a decrease in density in the case of products from different manufacturers. Despite the similarity of the board density and other parameters, most likely the recipe and the type of wood fibers used (from different species) had a greater impact on the obtained coating roughness.

The cited studies explain the observed inconsistency in the trend for roughness to increase with a decrease in density in the case of products from different manufacturers. Despite the similarity of the board density and other parameters, most likely the recipe and the type of wood fibers used (from different species) had a greater impact on the obtained coating roughness. Moreover, literature reports indicate that the density profile may be of greater importance for the shaping of the surface topography than the average density [34,68]. If a raw material with different parameters is used for the production of individual board layers, an uneven density profile can be expected. Manufacturers try to influence the shape of this profile by means of the parameters used in the process of their production (e.g., press closing speed, humidity, temperature, pressing time) [34,69,70]. The large variety of production technologies and types of panels causes differences in this profile. It was proved in this study that the average plate density is not a sufficient factor to enable determination of the roughness profile variability—especially when the samples come from two different manufacturers using different recipes and different pressing curves. This can be confirmed by literature data.

Sala published the results of research on the effect of the amount of aqueous solution of the release agent in different concentrations on the overheating of the fibrous carpet in the production of HDF boards, and the shaping of the density profile [71]. It was concluded that the sprayed amount of the solution has a significant impact on these parameters. In the range 0 to 32 mL/m^2^, a gradual increase (up to 5%) in the maximum and minimum density of the agent on the density profile was recorded. The continuation of the experiment proved that the amount of the applied solution and the temperature of the heating section of the press also affect the mechanical and physical properties [57]. In the case of surface roughness, a decrease of 31% was recorded. Any increase in the grain size of the sanding belt used to grind the HDF boards, in the P150–P400 range, resulted in a decrease in roughness values, with the exception of the Rku parameter. The surface profile parameters of the samples decreased gradually for each configuration as the grain size of the last sanding belt used increased. There was a clear decrease in roughness with the introduction of a P220 abrasive belt in addition to the P150 belt. A significantly smaller reduction in parameters was observed between the combination P150–P220 and the P220 belt. In the case of samples ground with one P220 abrasive belt in comparison with the combination P220–P320, a significant decrease in the indicators was recorded. The mean difference in values between the combination P220–P320 and the P400 belt was less than that between P150–P220 and P220. The final difference in roughness, between the configurations P400 and P400–P400, was markedly smaller than the previous differences between P150 and P150–P220, and P220 and P220–P320. The general tendency for a decrease in basic roughness parameters along with an increase in the grit of the sanding belt is confirmed by previously reported results for wood, particleboards, and fiberboards [39,72,73,74,75,76,77].

Figure 4 and Figure 5 show examples of values of the *Ra* parameters in the longitudinal and transverse directions, depending on the sanding program, feeding speed, and type of board.

Nemli et al., who examined particleboard, indicated an increase in the roughness parameters *Ra* and *Rz* with an increase in the feeding speed in the range 40–50–60 m/min [78]. Despite the lack of a statistically significant influence of this factor on the results of the present study, the obtained main effects plot also shows the tendency of the roughness parameters to increase when the conveyor belt feeding speed was increased from 25 to 50 m/min. The data obtained confirm the observations of other authors. Nemli points out that lower speed means longer machining time, during which dust removal and surface smoothing are more effective [78]. Previous studies investigating the correlation between cutting speed and feed speed per jag in the milling of HDF and MDF boards have shown that the roughness decreases with an increase in spindle speed and a decrease in feed per revolution. The authors of those studies also noted the large impact of the material removal rate on the obtained surface profile parameters [79,80]. The statistically insignificant, but observable influence of this factor on the results of the present study may imply that there was sufficient dust extraction (20 m/s). The total air requirement for two units of a longitudinal grinding and blowing conveyor belt was 130 m^3^/min.

The values of all roughness parameters measured along the grinding direction were lower than those measured across, except for the *Rk* parameter, which increased by 2.257 µm (31.9%). However, statistically significant differences occurred only for the amplitude parameters (the differences averaged 0.6 µm for *Ra* and 0.563 µm for *Rq*, representing a decrease by 17.8% and 12.4%, respectively). Hiziroglu et al. also reported such a relationship but did not find a statistically significant difference for any of the parameters *Ra*, *Rq*, and *Rz*. They only indicated that for *Ra,* the difference between the directions was 0.44 µm for particleboard and 0.19 µm for MDF [81]. On the other hand, when examining veneers, Li et al. recorded significant differences for the *Ra*, *Rsm*, *Rq*, and *Rz* parameters [82].

### 3.2. Optical Measurement Method

The numerical data obtained by the optical method were summarized in a table together with the values obtained by the contact method (Table 4 and Table 5). Assessing the numerical data and especially the generated images (Figure 6 and Figure 7) it was observed that the sanding process contributed to the effective formation of the sample surface.

When comparing the collected results, it was found that the values obtained with the profilometer were lower than the measurements made with the optical analyzer. The obtained results are largely confirmed by the work of Hazir and Koc, in which those authors, examining Lebanon cedar (Cedrus libani) and black pine (Pinus nigra) using a laser robotic measuring system and a pin-type meter, also obtained higher values with the non-contact method [83].

It is difficult to indicate unequivocally the reasons for the identified dependencies. It can be assumed that the reason for the discrepancy in the measurement results may be surface deformation caused by the detector pressure, which was 0.75 mN, resulting in underestimation of the results. Moreover, apart from the surface force generated by the stylus, the radius of the needle tip and the cut-off length of the profile also influence the values [33].

The results generally showed that the roughness parameters were subject to similar trends. The recorded data depended on the direction of measurement. For the tests performed in the longitudinal direction, lower roughness was usually noted [39,49]. It was also observed that the surface quality improves with the use of smaller abrasive grains in the second grinding step. This reflects the roughness of the substrate, which is reduced. This is confirmed both by the recorded numerical data and by the photographs of the surface topography, where a smaller proportion of red fields is observed. This is in line with the results of other studies [39,50,54,78,84,85].

In the measurements using the optical method, the ranges of values of the *Ra* parameter in the longitudinal direction after initial sanding with P150 paper at the speeds applied were 6.00–9.14 µm for board A, and 6.00–7.26 µm for board C. With the combination of two gradations on the sanding belt, the values decreased to 5.89–6.96 µm for board A and to 4.80–5.63 µm for board C. A greater reduction in the *Ra* parameter was achieved at a lower speed: for board A the reduction was 24% in the longitudinal direction and 30% in the transverse direction, while for board C the reductions were 33% and 27%, respectively. In the results obtained by the contact method, a different trend was observed: the value of the parameter showed a greater decrease at a speed of 50 m/min, by approximately 30% for board A and 20% for board C in the longitudinal direction and by approximately 35% for board A and 13% for board C in the transverse direction. The remaining roughness parameters recorded using the optical method showed similar trends to the *Ra* values, except for *Rv* and *Rz* for two-belt configurations and a speed of 50 m/min in the case of board A. Assessing the value of the *Ra* parameter in relation to *Rz* for the entire tested range of technological variables, it was found that the value of the *Ra* parameter was lower for board A by 8.66–9.96 in the longitudinal direction and by 7.45–12.22 µm in the transverse direction, while for board C it was lower by 8.44–10.23 µm and 9.77–11.14 µm, respectively. The values obtained are higher than those reported in other studies carried out on MDF boards with respect to wood, this being due to the homogeneous structure. The higher ratio of *Rz* to *Ra* may also be associated with the higher density of HDF boards, compared with MDF.

Considering the response of parameters to changes in the sanding speed, no unequivocal tendencies were shown. In the case of board A, a generally decreasing trend was noted in the longitudinal direction for both measuring methods with an increase in the sanding speed, except in the case of the P150 granulation treatment. For board C, such a tendency was found for the tests carried out using the optical method and for the P150/P220 program using the contact method. The same relationship was observed for board A in the transverse direction, while on the surface of board C at P150, such a relationship was obtained in optical measurements.

In these specified cases, the data obtained differ from the results of experiments reported by other authors, who found that when using the same granularity of abrasive belts and reducing the speed of the conveyor, the surface roughness decreases as a result of a longer impact of the abrasive belt on the treated surface [50,86]. Palija et al. showed that by using a higher conveyor speed and less granular sanding belts in the final sanding stage, the best results can be obtained without negatively affecting the quality of the machining [54]. According to the authors, this solution should be used when preparing MDF boards for further production stages.

When assessing the generated surface topography images, it was found that the changes in the amplitude of the roughness were greater after the treatment with P150 paper. The use of two combinations of grains resulted in a reduction in the roughness, which had a positive effect on the appearance. Compared with the samples before grinding, on which there were clearly visible bundles of fibers pressed in the surface layer and free spaces between them that were observed, the other variants exhibited greater homogeneity. The microscopic photos of the surface after sanding show that the structure of the fiber bundles is broken, which makes them shorter. Moreover, for samples made from board C, despite the lower density, generally lower values of roughness parameters were recorded, which is in line with the results of Akbulut and Koç [34,56]. On the other hand, a study by Hiziroglu did not show an unequivocal influence of density on the obtained values [18]. These differences may be explained by the properties of the raw materials used to produce the boards used for the tests (type of fibers, introduced additives), the heat treatment of the fiber, and pressing parameters [56,87,88,89].

### 3.3. Gloss Level

The results of the gloss measurements for the selected variants are presented in the form of arithmetic means in Figure 8.

The measurement results showed that the values for the sanded board were lower by an average of 2.7 times than those for the twin samples of raw board. The lower roughness of the sanded surface of the HDF board, which was documented on the images in optical measurements, contributed to the reduction in the measurement data. The introduction of two tape configurations did not lead to any significant changes. According to the criteria given in the table, the tested variants were assessed as matt. For the control samples, the higher recorded values may have been caused by the layer of agents improving the hydrophobicity and anti-adhesiveness, applied by the producers to the surface of the boards [15]. All agents were removed when the surface of the board was sanded.

Comparing the test results concerning the gloss of UV varnish coatings formed on the same types of boards, sanded with P150 and P220 abrasive belts at a speed of 50 m/min, it was found that the gloss value was significantly higher than that of the raw panels. On the other hand, the measurements of the roughness of the lacquer coatings showed that the gloss increased with a decrease in the values of the recorded parameters on their surface. This trend was different than expected. As other reports indicate, the final gloss effect is influenced not only by the substrate but also by the properties of the applied varnish products, as well as the methods of their application or hardening [44,90]. Nevertheless, regardless of the roughness of the boards, the gloss of the coatings was within the range corresponding to a semi-matt finish.

## 4. Discussion and Conclusions

The measurements showed that both the contact and non-contact systems may be used to check the roughness parameters, despite the recorded differences. They can supply objective data and may constitute a criterion for the assessment of the surface condition, determining the proper performance of technological operations in the further stages of finishing with the use of various products. Possibilities of using these methods have also been signaled in the literature [35,83].

The contact method is more widely used in production conditions for product quality control. By checking the roughness parameters of the products, board manufacturers can control the surface quality. They can influence the manufacturing technology, taking into account both material and technological parameters. One of the most important properties of the boards is the density profile. On the other hand, furniture manufacturers should pay attention to the selection of sanding parameters. In the case of finishing of boards with varnish products or thin cladding materials, where the surfaces must be very smooth, this is a particularly important issue. The determination of the properties of materials after the finishing process is a delayed action. Any errors occurring at the production stage are a source of defects that cause the product no longer to meet customer expectations. For businesses, these bring not only damage to image but also financial losses. Such problems are particularly acute in the case of large-scale production.

The importance of these issues is indicated by the high activity of scientists in research on the surface roughness of both wood and boards. The literature on this subject describes the material and technological issues influencing the roughness. Due to the variability of the quality of the raw material and the production parameters of the boards, the surface roughness should be checked before the board is sent for finishing. The data collected in this work may provide a basis for assessing the quality of HDF boards before finishing, which affects the aesthetic and decorative values of the final product.

Based on the experimental results and theoretical considerations, the following conclusions were drawn:Density is not a determining factor for the surface roughness of HDF boards. HDF boards produced by various manufacturers using fibers of different origins and degrees of fragmentation presented different roughness profiles despite the similar physical and mechanical properties of the finished boards. Among the samples from a single manufacturer, an increase in roughness with a decrease in density was observed.Based on analysis of variance (ANOVA) performed for the parameters *Ra*, *Rq*, *Rku*, *Rz*, *Rp*, and *Rv*, it was determined that the type of board used (except in the case of *Rku* and *Rsk*) and the configurations of the sanding belts (except in the case of *Rv*) had a statistically significant impact on the roughness. Additionally, the direction of measurement of roughness had a significant influence on the amplitude parameters (*Ra*, *Rq*, *Rz*).There was found to be a tendency for the HDF surface roughness to decrease with an increase in the grain size of the abrasive paper used in the grinding process.The feeding speed of the conveyor belt did not have a significant effect on the obtained roughness with the dust extraction in the sanding machine equal to 20 m/s.The results of roughness measurements carried out on the samples in the grinding direction were lower than those obtained in the transverse direction.In the roughness measurements by the optical method, higher values were obtained while maintaining similar trends as the contact method.The gloss values of the tested boards, regardless of the sanding program used, lay within the range corresponding to a matt finish.

## Figures and Tables

**Figure 1 materials-15-06359-f001:**
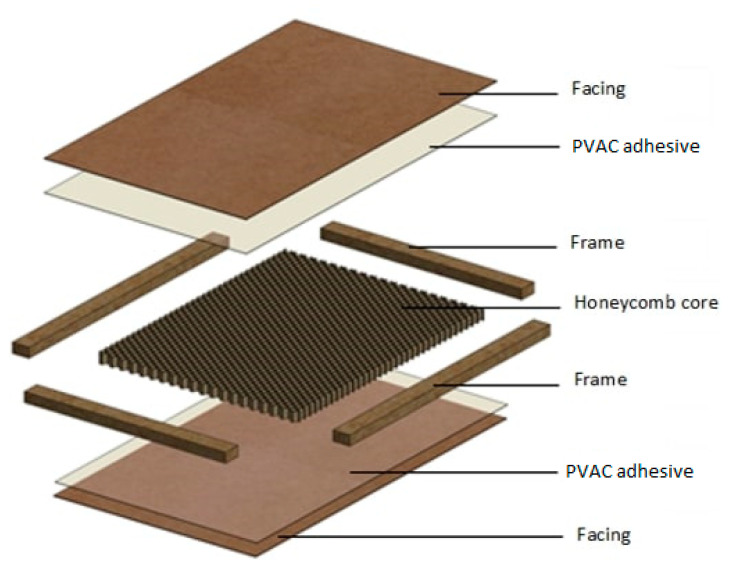
Construction of board on frame with honeycomb core.

**Figure 2 materials-15-06359-f002:**
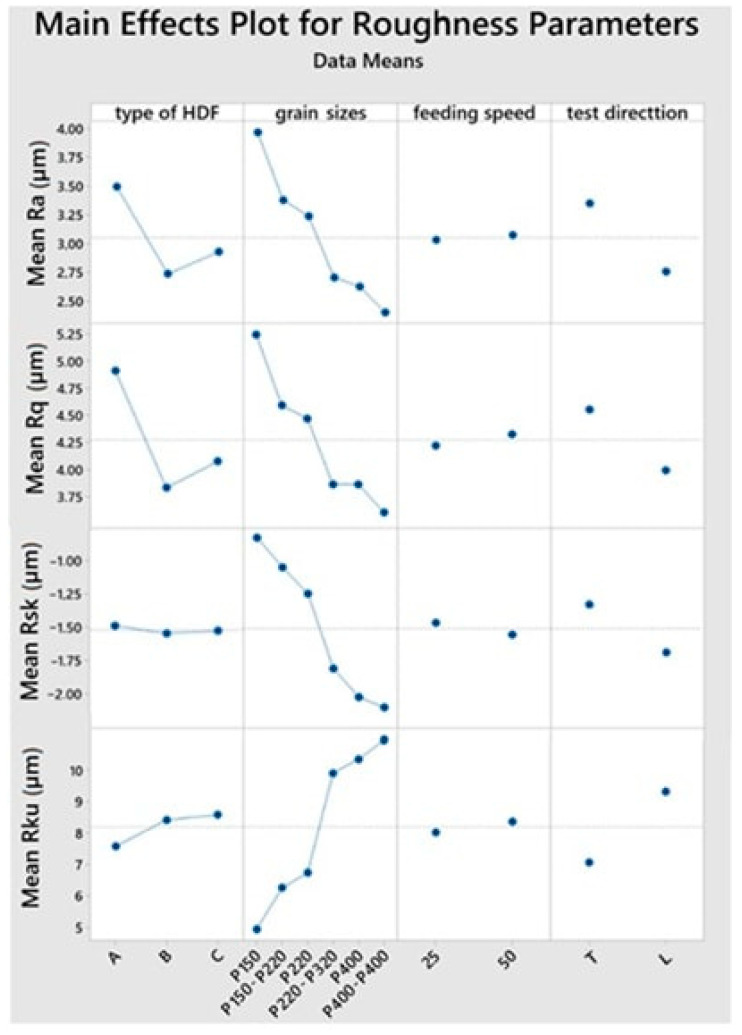
Main effects plots for mean of amplitude roughness parameters (arithmetic mean deviation *Ra*, geometric average roughness *Rq*, kurtosis of the roughness profile *Rku*, skewness of the roughness profile *Rsk*) for four factors.

**Figure 3 materials-15-06359-f003:**
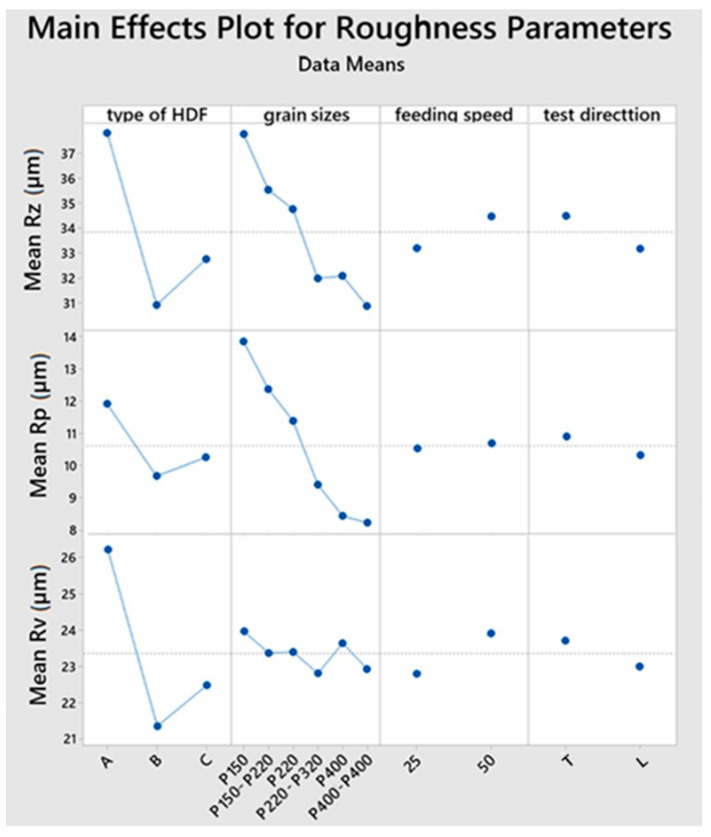
Main effects plots for mean of height roughness parameters (the maximum peak height of the roughness profile *Rp*, the maximum valley depth of the roughness profile *Rv*, ten-point height *Rz*) for four factors.

**Figure 4 materials-15-06359-f004:**
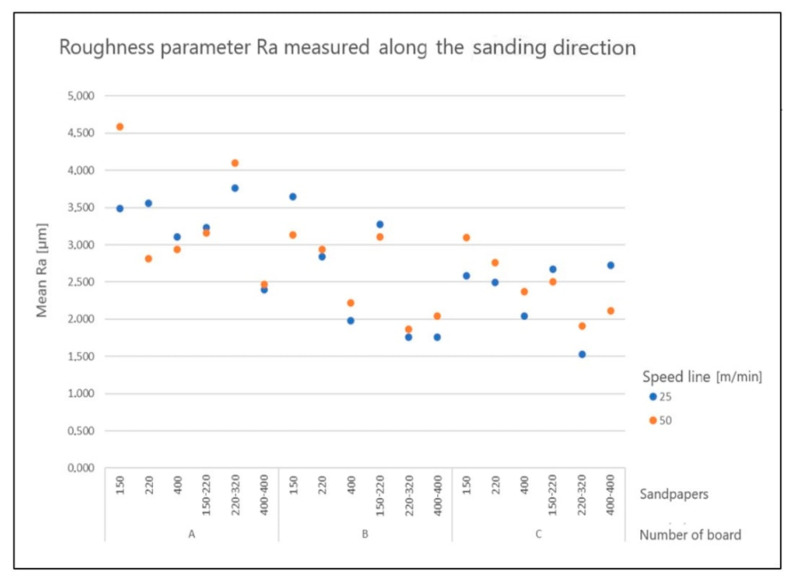
Structure of arithmetic mean deviation *Ra* in the longitudinal direction, depending on the sanding program.

**Figure 5 materials-15-06359-f005:**
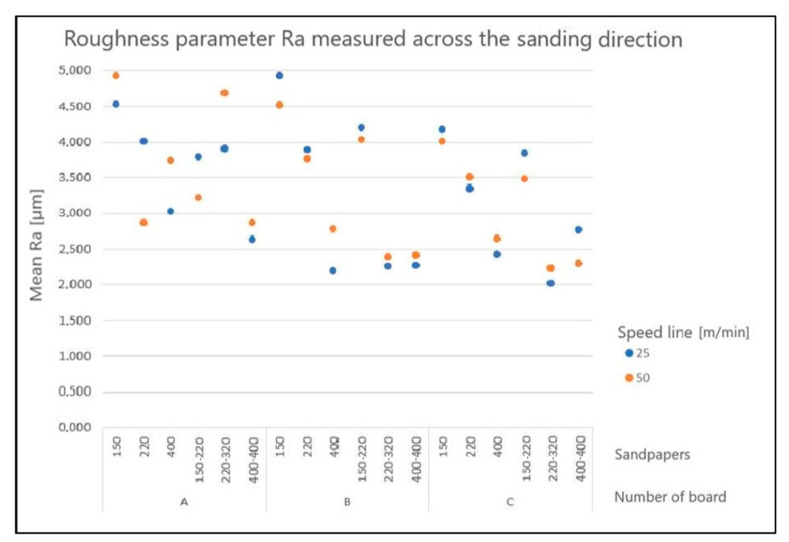
Structure of arithmetic mean deviation *Ra* in the transverse direction, depending on the sanding program.

**Figure 6 materials-15-06359-f006:**
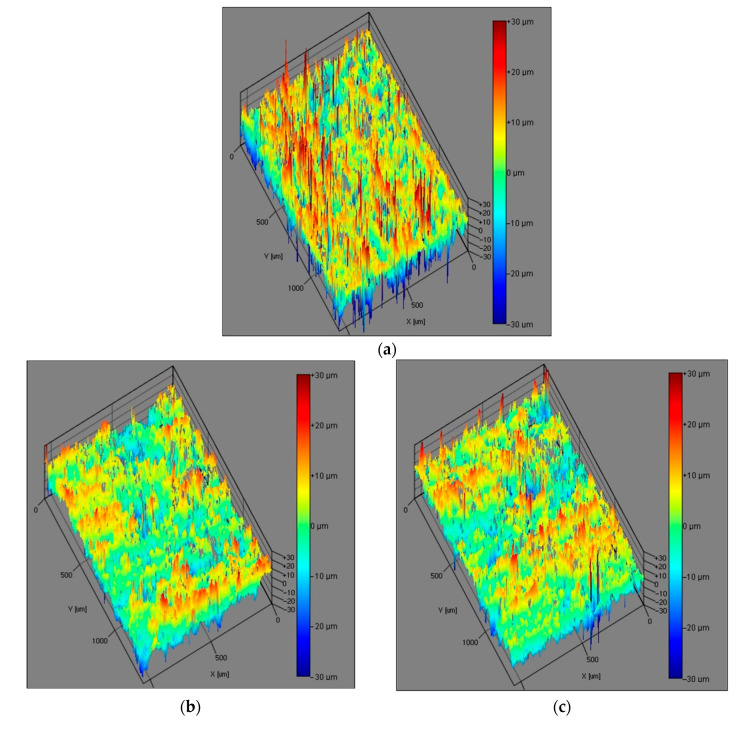
Surface topography of the board C (**a**) before sanding (**b**) sanding program P150, feeding speed 25 m/min (**c**) sanding program P150, feeding speed 50 m/min (**d**) sanding program P150/P220, feeding speed 25 m/min (**e**) sanding program P150/P220, feeding speed 50 m/min.

**Figure 7 materials-15-06359-f007:**
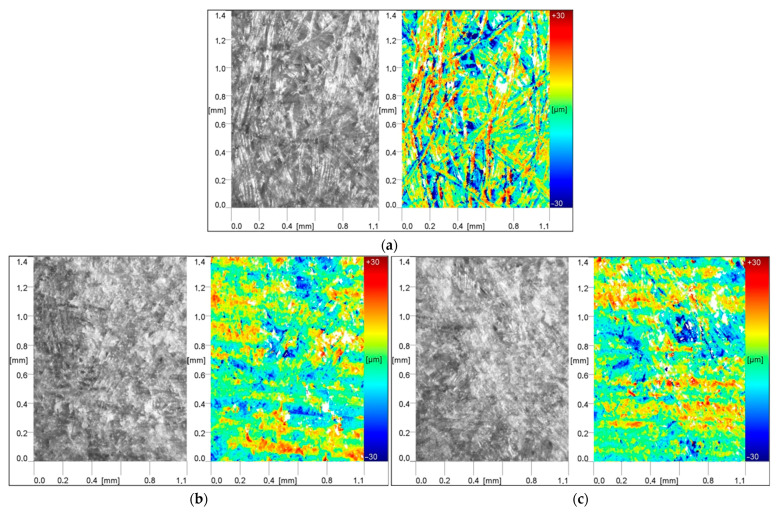
View of the surface of the C board (**a**) before sanding (**b**) sanding program P150, feeding speed 25 m/min (**c**) sanding program P150, feeding speed 50 m/min (**d**) sanding program P150/P220, feeding speed 25 m/min (**e**) sanding program P150/P220, feeding speed 50 m/min.

**Figure 8 materials-15-06359-f008:**
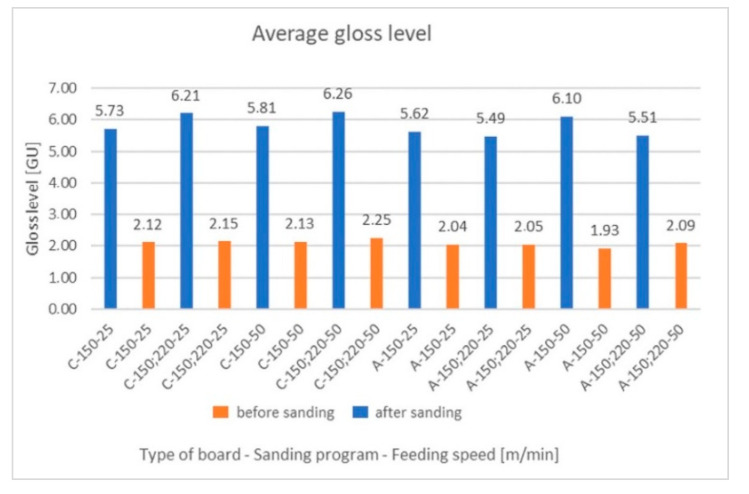
Average gloss level before and after sanding board A and C.

**Table 1 materials-15-06359-t001:** Basic information on the boards used.

Board Label	Supplier	Modulus of Rupture (N/mm^2^)	Modulus of Elasticity (N/mm^2^)	Swelling after 24 h (%)	Density (kg/m^3^) acc. to PN-EN 323:1999
A	1	>38	>3400	<60	850
B	2	>45	4300	35	850
C	2	>45	4300	45	830

**Table 2 materials-15-06359-t002:** Technical specifications 3D Topography module with OneAttension Theta.

Method	Fringe Projection Phase-Shifting
XY pixel size:	1.1 μm × 1.1 μm
Measured range in Z direction	1–60 μm
Lateral sampling (XY):	1.41 mm × 1.06 mm
Measurement speed	5–30 s (1280 × 960 measurement points)
Imaging options	Optical image, 2D and 3D roughness graphs

**Table 3 materials-15-06359-t003:** One-way ANOVA analysis of roughness parameters (arithmetic mean deviation *Ra*, geometric average roughness *Rq*, kurtosis of the roughness profile *Rku*, skewness of the roughness profile *Rsk*, the maximum peak height of the roughness profile *Rp*, the maximum valley depth of the roughness profile *Rv*, ten-point height *Rz*) as a function of the variables.

One-Way ANOVA Response	Roughness Parameter	DF The Total Degrees of Freedom	Adj SS Adjusted Sums of Squares	Adj MS Adjusted Mean Squares	F-Value	*p*-Value
	*Ra*	2	7.493	3.746	6.080	0.004
	*Rq*	2	15.410	7.707	10.520	0.000
	*Rz*	2	614.600	307.290	17.580	0.000
Type of HDF	*Rp*	2	65.110	32.553	4.710	0.012
board	*Rv*	2	312.000	156.004	36.190	0.000
	*Rsk* *(row data)*	2	0.040	0.020	0.050	0.947
	*Rsk (johnson transformation data)*	2	0.113	0.057	0.050	0.949
	*Rku*	2	14.000	6.999	0.750	0.475
	*Ra*	5	20.550	4.110	9.220	0.000
	*Rq*	5	22.440	4.489	6.810	0.000
	*Rz*	5	415.500	83.090	3.900	0.004
Grain size of sanding belts	*Rp*	5	313.600	62.719	18.110	0.000
	*Rv*	5	11.310	2.262	0.250	0.939
	*Rsk* *(row data)*	5	17.682	3.536	32.410	0.000
	*Rsk (johnson transformation data)*	5	46.840	9.367	22.750	0.000
	*Rku*	5	381.000	76.206	18.310	0.000
	*Ra*	1	0.030	0.030	0.040	0.839
	*Rq*	1	0.182	0.182	0.190	0.662
	*Rz*	1	29.700	29.700	1.160	0.285
Feeding speed	*Rp*	1	0.494	0.494	0.060	0.801
	*Rv*	1	22.290	22.294	2.660	0.108
	*Rsk* *(row data)*	1	0.148	0.148	0.420	0.519
	*Rsk (johnson transformation data)*	1	0.432	0.432	0.410	0.523
	*Rku*	1	2.233	2.233	0.240	0.626
	*Ra*	1	6.387	6.387	10.260	0.002
	*Rq*	1	5.703	5.703	6.620	0.012
Direction of the	*Rz*	1	31.070	31.070	1.220	0.274
roughness	*Rp*	1	5.961	5.961	0.780	0.381
measurement	*Rv*	1	9.145	9.145	1.070	0.305
	*Rsk* *(row data)*	1	2.378	2.378	7.400	0.008
	*Rsk (johnson transformation data)*	1	9.898	9.897	10.810	0.002
	*Rku*	1	91.700	91.697	11.380	0.001

**Table 4 materials-15-06359-t004:** Roughness parameters for the A board.

Parameter	Grain Sizes Feeding Speed							
	P150 25 m/min		P150 50 m/min		P150/220 25 m/min		P150/220 50 m/min	
	Measurement Method							
	Optical	Profilometer	Optical	Profilometer	Optical	Profilometer	Optical	Profilometer
	Horizontal							
*Ra* [μm]	9.135	3.485	6.000	4.584	6.956	3.237	5.890	3.157
*Rq* [μm]	11.899	4.886	8.032	6.250	8.973	4.664	7.818	4.439
*Rp* [μm]	37.511	13.876	25.289	15.453	35.920	12.367	29.205	14.993
*Rv* [μm]	−43.859	24.115	−27.495	28.707	−31.417	26.225	−29.444	23.247
*Rz* [μm]	81.370	37.691	52.784	43.910	67.337	37.891	58.649	38.240
	**Vertical**							
*Ra* [μm]	8.185	4.523	7.834	4.919	5.685	3.787	6.035	3.209
*Rq* [μm]	11.542	5.829	9.873	6.423	8.071	5.008	8.027	4.645
*Rp* [μm]	49.138	14.308	29.170	15.640	24.979	12.036	25.688	13.577
*Rv* [μm]	−50.893	26.154	−31.503	29.676	−35.360	24.713	−30.340	26.237
*Rz* [μm]	100.031	40.162	60.674	45.315	60.340	36.749	56.029	37.614

**Table 5 materials-15-06359-t005:** Roughness parameters for the C board.

Parameter	Grain Sizes Feeding Speed							
	P150 25 m/min		P150 50 m/min		P150/220 25 m/min		P150/220 50 m/min	
	Measurement Method							
	Optical	Profilometer	Optical	Profilometer	Optical	Profilometer	Optical	Profilometer
	Horizontal							
*Ra* [μm]	7.262	2.583	6.005	3.098	4.797	2.673	5.630	2.505
*Rq* [μm]	9.464	3.582	7.742	4.265	6.434	3.745	7.165	3.518
*Rp* [μm]	31.744	10.026	27.550	10.835	20.781	10.662	17.794	9.501
*Rv* [μm]	−42.623	19.265	−30.990	21.298	−27.840	21.046	−26.661	19.749
*Rz* [μm]	74.367	29.291	58.540	32.132	48.620	31.707	44.454	28.699
	**Vertical**							
*Ra* [μm]	7.162	4.165	7.821	4.006	5.245	3.842	6.373	3.481
*Rq* [μm]	9.558	5.229	10.272	5.160	7.131	4.884	8.844	4.529
*Rp* [μm]	27.398	13.424	42.307	13.599	26.575	11.363	40.843	11.412
*Rv* [μm]	−42.577	20.321	−35.235	22.764	−31.862	21.651	−41.996	20.972
*Rz* [μm]	69.975	33.745	77.542	36.363	58.437	32.814	82.839	32.530

## Data Availability

Department of Wood Science and Thermal Techniques, Faculty of Forestry and Wood Technology, Poznan University of Life Sciences, Wojska Polskiego St. 38/42, 60-627 Poznan, Poland; milena.henke@up.poznan.pl.

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
