# Peer review of "Evaluation of Surface Roughness Parameters of HDF for Finishing under Industrial Conditions"

_materials, 2022, doi:10.3390/ma15186359_

Round 1
Reviewer 1 Report
This is an interesting study on surface roughness on a wood composite material. The results add to the field, however I have reservations on the statistics that need to be addressed before the paper can be published.
How was it determined where the differences lay within the ANOVA?
Without a normal distribution, the results of the ANOVA are invalid. The data should have been transformed into a normal distribution before running the ANOVA. A quick transformation and rerunning of the data should be done, and the paper changed accordingly.
Author Response
Thank you very much for your kind reminder. The changes to the main text are marked yellow. A point-by-point response to the raised questions is shown below:
- This is an interesting study on surface roughness on a wood composite material. The results add to the field, however I have reservations on the statistics that need to be addressed before the paper can be published.
Answer:
Thank you very much for your positive comment. We have carefully checked the statistics and carried out the analysis with the person responsible for the statistical analysis.
- How was it determined where the differences lay within the ANOVA?
- Without a normal distribution, the results of the ANOVA are invalid. The data should have been transformed into a normal distribution before running the ANOVA. A quick transformation and rerunning of the data should be done, and the paper changed accordingly.
Answer:
The normality of the distribution for all the results was verified, the data did not have a normal distribution for the parameter Rsk (p-value = 0.034), for all other p-value level is > 0.05. It was confirmed that there was no significant interference with the experiment. A Johnson transformation was performed for Rsk parameter. Anova was performed for this data. The difference between the influences of different parameter settings is on a similar level after the transformation. Overall conclusions remain the same.
Please see attachement (a figure).

Reviewer 2 Report
This work was to determine the effect of various sanding belt configurations and the feeding speed of the conveyor belt during grinding on the surface roughness of high-density fiberboards (HDF).
It can be considered for acceptance for publication in the journal after major revision.
1. What is the novelty in this work? It should be clearly highlighted.
2. There are several grammatical mistakes throughout the manuscript, correct it.
Author Response
Thank you very much for your kind reminder. A point-by-point response to the raised questions is shown below. If you have additional comments regarding the revised manuscript, please let us know.
- This work was to determine the effect of various sanding belt configurations and the feeding speed of the conveyor belt during grinding on the surface roughness of high-density fiberboards (HDF).
It can be considered for acceptance for publication in the journal after major revision.
Answer:
Thank you very much for your comment. We have carefully checked the text and made some additions. The English language has been checked by a native-speaker from the official office.
- What is the novelty in this work? It should be clearly highlighted.
Answer:
This is preliminary research as part of a larger project. The novelty of this research is the surface finishing of high density fibreboard (HDF) with a UV varnish system using energy efficient energy sources. Research on substrate preparation, carried out under industrial conditions, is of great importance in further stages of the finishing process and affects the final quality of coatings.
In general, there is little dissemination of this data by companies due to the fact that each manufacturer develops a sanding-to-paint system on the basis of its own experience, which is the source of creating a competitive advantage.
- There are several grammatical mistakes throughout the manuscript, correct it.
Answer:
A linguistic correction has been carried out. English was checked by a native-speaker from the official office.
Please see attachement.

Reviewer 3 Report
The authors need to make small changes:
- what is the meaning of the mention in line 167;
- in the introduction to present in more detail how the surface roughness influences the properties of the finished product;
- to emphasize the novelty of the research.
Author Response
Thank you very much for your kind review. The changes to the main text are marked yellow. A point-by-point response to the raised questions is shown below. If you have additional comments regarding the revised manuscript, please let us know.
The authors need to make small changes:
1. What is the meaning of the mention in line 167;
Answer:
Thank you the reviewer - our mistake. The text „in Fig. 2 & 3: great-> grain“ was removed.
- In the introduction to present in more detail how the surface roughness influences the properties of the finished product;
Answer:
In the production of commercial wood or wood-based functional components, the quality of the finishings is very important, which depends, among other things, on the smoothness of the substrate (Ozdemir et al. 2015). It is therefore necessary to know the topography of the substrate to be finished. With this in mind, it is important for furniture technologists to know the roughness and how to shape it. By being able to select the type of abrasive and granulation during processing, it is possible to bring about the intended change in profile, while also influencing the wear of the applied lacquer products, the technical and optical properties of the lacquer coatings obtained from them (Zhao-Wei 2021).
- To emphasize the novelty of the research.
Answer:
This is preliminary research as part of a larger project. The novelty of this research is the surface finishing of high density fiberboard (HDF) with a UV varnish system using energy efficient energy sources. Research on substrate preparation, carried out under industrial conditions, is of great importance in further stages of the finishing process and affects the final quality of coatings.
In general, there is little dissemination of this data by companies due to the fact that each manufacturer develops a sanding-to-paint system on the basis of its own experience, which is the source of creating a competitive advantage.

Reviewer 4 Report
This paper presents an evaluation of different surface roughness parameters (Ra, Rq, Rku, Rsk, Rp, Rv, Rz) for HDF boards using statistical analysis by one-way ANOVA. The document is well-written and fits in the scope of Materials. However, the data discussion and conclusions need improvement to demonstrate scientific soundness and differentiate it from a technical report. Please find below some major and minor comments:
Major comments:
- The data discussion lies around the results of the statistical analysis, which is based on a limited number of experimental conditions and with a limited physical interpretation of the process phenomena which hinders the paper’s aims of providing valuable guidance for the furniture industry. I would recommend improving the discussion section by adding the physical interpretation of the process and highlighting what are the scientific breakthroughs of this paper.
- In my opinion, a full explanation of the different roughness parameters (Ra, Rq, Rku, Rsk, Rp, Rv, Rz) should be given in this paper rather than only referenced. A critical discussion about which parameter should be used for process evaluation and which is potentially important for the identification of HDF roughness issues would be also useful.
Minor comments:
Lines 48 to 50 – Please define HDF, MDF and Ra in the text.
Line 73 – I would recommend changing “Salca et al.” to “Some authors”.
Line 88 – I would use weight per area rather than weight.
Line 92 – The correct decimal place should be 42.0.
Line 154 - Define DF, SS and MS in Table 2.
Line 167 – What the comment “[in Fig. 2 & 3: great-> grain]” mean?
Author Response
Thank you very much for your kind review. We have now carefully addressed all points in question. The changes to the main text are marked yellow. A point-by-point response to the raised questions is shown below. If you have additional comments regarding the revised manuscript, please let us know.
This paper presents an evaluation of different surface roughness parameters (Ra, Rq, Rku, Rsk, Rp, Rv, Rz) for HDF boards using statistical analysis by one-way ANOVA. The document is well-written and fits in the scope of Materials. However, the data discussion and conclusions need improvement to demonstrate scientific soundness and differentiate it from a technical report. Please find below some major and minor comments:
Major comments:
- The data discussion lies around the results of the statistical analysis, which is based on a limited number of experimental conditions and with a limited physical interpretation of the process phenomena which hinders the paper’s aims of providing valuable guidance for the furniture industry. I would recommend improving the discussion section by adding the physical interpretation of the process and highlighting what are the scientific breakthroughs of this paper.
Answer:
Lacquer coatings manufacturers do not provide any guidelines regarding the roughness profile of the surface to be treated. In the production of commercial wood or wood-based composites functional parts, the quality of the coatings is very important. It depends, among other things, on the smoothness of the substrate (Ozdemir et al. 2015). Each company develops its own technological parameters e.g. roughness.
It is therefore necessary to know the topography of the finished substrate. It is important for furniture technologists to know the roughness of surfaces in aspects of finishing technology. Thanks to the possibility of selecting the type of abrasive and granulation during processing, it is possible to bring about an intended change in the profile while at the same time influencing the consumption of applied lacquer products, technical and optical properties of the lacquer coatings obtained from them (Zhao-Wei 2021). Along with the development of the furniture industry, as well as advanced technologies used in the refining process, there is also an increase in the requirements for finishes, which should fulfil not only a protective but also an aesthetic function.
In the next step of investigations we will use different analysis methods for the testing of substrate and final coatings.
- In my opinion, a full explanation of the different roughness parameters (Ra, Rq, Rku, Rsk, Rp, Rv, Rz) should be given in this paper rather than only referenced. A critical discussion about which parameter should be used for process evaluation and which is potentially important for the identification of HDF roughness issues would be also useful.
Answer:
Individual standards do not recommend which roughness parameters should be taken into account when considering the geometric structure of the surface.
During finishing are taking into account, the roughness parameters that determine the height of profile irregularities are most often taken into account.
The amount of applied lacquer products must be chosen so that the coating covers any surface irregularities.
Minor comments:
- Lines 48 to 50 – Please define HDF, MDF and Ra in the text.
Answer:
HDF, MDF and Ra terms were defined in the text.
- Line 73 – I would recommend changing “Salca et al.” to “Some authors”. Answer:
“Salca et al.” into “Some authors” was changed.
- Line 88 – I would use weight per area rather than weight.
Answer:
Accordingly tot he reviewer suggestion the term „weight per area“ was used.
- Line 92 – The correct decimal place should be 42.0.
Answer:
Correct decimal place was used.
- Line 154 - Define DF, SS and MS in Table 2.
Answer:
DF, SS and MS were defined in Table 2.
- Line 167 – What the comment “[in Fig. 2 & 3: great-> grain]” mean?
Answer:
Thank you the reviewer - our mistake. The text „in Fig. 2 & 3: great-> grain“ was removed.

Reviewer 5 Report
Comments to authors
In this manuscript, the authors evaluated the surface roughness parameters of HDF for finishing under industrial conditions. In general, this manuscript is more like technical analysis to obtain the parameters which affect the surface roughness. No specific scientific issue was proposed and then resolved. Thus, a major revision is highly recommended. Some comments were shown as follows:
1-Some tests, such as SEM and AFM, are highly needed to show the varied roughness of boards.
2-More information about the physicochemical composition of HDF is needed. It could also be related to the roughness of the board during the production.
3-For the feeding speed and test direction, two points are not enough to accurately show the trend. At least three points are recommended.
Author Response
Thank you very much for your kind review. The changes to the main text are marked yellow. A point-by-point response to the raised questions is shown below. If you have additional comments regarding the revised manuscript, please let us know.
In this manuscript, the authors evaluated the surface roughness parameters of HDF for finishing under industrial conditions. In general, this manuscript is more like technical analysis to obtain the parameters which affect the surface roughness. No specific scientific issue was proposed and then resolved. Thus, a major revision is highly recommended. Some comments were shown as follows:
- Some tests, such as SEM and AFM, are highly needed to show the varied roughness of boards.
Answer:
Thank you very much for a very good advice. There are preliminary research as part of a larger project. The novelty of this research is the surface finishing of high density fibreboard (HDF) with a UV varnish system using energy efficient energy sources. In the next step of investigations we will use different analysis methods for the testing of substrate and final coatings.
Roughness is a very important in terms of the varnishing of lignocellulosic substrates. We agree with the reviewer that the surface morphology should be evaluated in the micro level, which would extend the roughness tests. The presented paper focuses on the effect of surface roughness on the change of topography without going into the structure and chemical composition.
- More information about the physicochemical composition of HDF is needed. It could also be related to the roughness of the board during the production.
Thank you very much for a very good advice. We should mentioned that investigations in the industrial conditions were carried out. One Company would like to introduce into practice the new technology of finishing of HDF with a UV varnish system using energy efficient energy sources. We strongly agree with the reviewer. In the next papers we will try investigate composition of wood based composites into roughness parameters.
- For the feeding speed and test direction, two points are not enough to accurately show the trend. At least three points are recommended.
Thank you very much for a very good advice. We agree with the reviewer. There are results of preliminary investigations. In the next step of research we will use more parameters (e.g. feeding speed).
For deeper analysis of correlation between kind of substrate, sanding program (size grain), feeding speed and direction of measurement were estimated. Obtained results will be helpfull for prediction finishing parameters of HDF in the industrial conditions.

Round 2
Reviewer 2 Report
After revision, the manuscript can be accepted for publication.
Author Response
Answer:
Thank you very much for your positive review. The English language has been checked once again by a native-speaker from the official office. We made some additions and changes into English language text.

Reviewer 4 Report
The authors have improved the paper, however, additional data discussion and scientific interpretation of the process were not provided, hence the scientific soundness is low-to-average. It is claimed that basic information on HDF roughness and surface varnishing process is still lacking in the literature and this paper is an initial study in the field providing technical information. The authors mention that more in-depth analysis with different measurement techniques will be investigated in future papers. On this basis, I would recommend this paper for publication in Materials in the current form.
Author Response
The authors have improved the paper, however, additional data discussion and scientific interpretation of the process were not provided, hence the scientific soundness is low-to-average. It is claimed that basic information on HDF roughness and surface varnishing process is still lacking in the literature and this paper is an initial study in the field providing technical information. The authors mention that more in-depth analysis with different measurement techniques will be investigated in future papers. On this basis, I would recommend this paper for publication in Materials in the current form.
Answer:
Thank you very much for the very positive review and understanding our idea. The basic information related to the HDF as a substrate were given into the text.

Reviewer 5 Report
Although the authors have explained that this work was part of an industrial project, I still can not recommend publishing it without resolving the suggestions from the first round review. The results in the manuscript mainly discussed some phenomena rather than a scientific issue. The roughness was affected by several parameters, then why and how?
Author Response
Although the authors have explained that this work was part of an industrial project, I still can not recommend publishing it without resolving the suggestions from the first round review. The results in the manuscript mainly discussed some phenomena rather than a scientific issue. The roughness was affected by several parameters, then why and how?
Answers:
Thank you very much the reviewer for the second review. We checked carefully the text and made some changes and additions to the text.
- We added more information related to the properties of HDF used in investigations.
- We strongly agree with the reviewer that SEM and AFM analysis are very useful tools for deeper analysis of surfaces.
- Accordingly to the reviewer suggestions some sentences were added to the text.
- The surface roughness profile is more influenced by the density profile than the average density. Depending on what the density curve is, the roughness will be different. The near-surface density can different from the determined average density. It is most influenced by the pressing cycle. The appropriate setting of the pressing parameters at successive intervals, the so-called pressing process curve, influences the density profile. In this study, it was shown that the average density of a HDF is not a sufficient factor to be able to determine the variability of the roughness profile. Especially when the samples come from two different companies with different formulations and different pressing curves.
- Another factor that differentiates the roughness of a board is the release agent sprayed during pressing (between pre-pressing and actual pressing). Its main purpose is to ensure the free flow of the sheet, with no adhesion/bonding of the material at the various stages of production to the line elements. The interaction between release agent (agent quality, concentration used) and fiber moisture also influences the surface roughness profile.
- The English language has been checked by a native-speaker from the official office. We made some additions and changes into English language text (mostly in the Introduction section).
